# Safety of Biologic-DMARDs in Rheumatic Musculoskeletal Disorders: A Population-Based Study over the First Two Waves of COVID-19 Outbreak

**DOI:** 10.3390/v14071462

**Published:** 2022-07-01

**Authors:** Arianna Sonaglia, Rosanna Comoretto, Enrico Pasut, Elena Treppo, Giulia Del Frate, Donatella Colatutto, Alen Zabotti, Salvatore De Vita, Luca Quartuccio

**Affiliations:** 1Division of Rheumatology, Department of Specialist Medicine, Azienda Sanitaria Universitaria del Friuli Centrale (ASUFC), 33100 Udine, Italy; arianna.sonaglia23@gmail.com (A.S.); treppo.elena@gmail.com (E.T.); giulia.delfrate@gmail.com (G.D.F.); donatella.colatutto@gmail.com (D.C.); alen.zabotti@asufc.sanita.fvg.it (A.Z.); salvatore.devita@uniud.it (S.D.V.); 2Department of Medicine (DAME), University of Udine, 33100 Udine, Italy; 3Zeta Research S.r.l., 34133 Trieste, Italy; rosannacomoretto@zetaresearch.com; 4Service of Pharmacy, Azienda Sanitaria Universitaria del Friuli Centrale (ASUFC), 33100 Udine, Italy; enrico.pasut@asufc.sanita.fvg.it

**Keywords:** COVID-19, safety, rheumatic, autoimmune, outbreak, therapy

## Abstract

This study aims to explore disease patterns of coronavirus disease (COVID-19) in patients with rheumatic musculoskeletal disorders (RMD) treated with immunosuppressive drugs in comparison with the general population. The observational study considered a cohort of RMD patients treated with biologic drugs or small molecules from September 2019 to November 2020 in the province of Udine, Italy. Data include the assessment of both pandemic waves until the start of the vaccination, between February 2020 and April 2020 (first), and between September 2020 and November 2020 (second). COVID-19 prevalence in 1051 patients was 3.5% without significant differences compared to the general population, and the course of infection was generally benign with 2.6% mortality. A small percentage of COVID-19 positive subjects were treated with low doses of steroids (8%). The most used treatments were represented by anti-TNF agents (65%) and anti-IL17/23 agents (16%). More than two-thirds of patients reported fever, while gastro-intestinal symptoms were recorded in 27% of patients and this clinical involvement was associated with longer swab positivity. The prevalence of COVID-19 in RMD patients has been confirmed as low in both waves. The benign course of COVID-19 in our patients may be linked to the very low number of chronic corticosteroids used and the possible protective effect of anti-TNF agents, which were the main class of biologics herein employed. Gastro-intestinal symptoms might be a predictor of viral persistence in immunosuppressed patients. This finding could be useful to identify earlier COVID-19 carriers with uncommon symptoms, eventually eligible for antiviral drugs.

## 1. Introduction

In December 2019, a novel viral infection spread in Wuhan, a province of China, caused by a member of the Coronaviridae family, known as severe acute respiratory syndrome coronavirus 2 (SARS-CoV-2). The related disease, named COVID-19, rapidly diffused worldwide, so the World Health Organization declared it a Public Health Emergency of International Concern on January 2020 and a pandemic on 11 March 2020 [1,2,3]. The infected population presented a wide spectrum of manifestations, ranging from low-grade fever and mild respiratory symptoms to acute respiratory failure, and some people were even asymptomatic, making an early diagnosis difficult and facilitating virus circulation [1]. Moreover, the infection shows not only a wide range of manifestations, but also different degrees of severity, and therefore, different patterns of recovery [2,4]. The duration of manifestations and viral clearance from the host present large variability, from a few days to months. In fact, some people completely recover in a short period while others can show persistent manifestations as chronic fatigue, muscle weakness, sleeping difficulties, anxiety and depression [5,6,7]. Finally, some patients develop several problems due to non-reversible organ damage, such as pulmonary fibrosis, renal failure and cardiac problems [1,3].

In predisposed individuals, COVID-19 can induce Interleukin-6 (IL-6) dependent pathways related to cytokine storm and macrophage activation syndrome (MAS) [8,9,10,11,12,13,14]. Moreover, SARS-CoV-2 infection might alter interferon signaling and activate other inflammatory mechanisms. Based on this evidence, immunomodulatory therapies have been employed to treat selected cases of severe COVID-19. Therefore, immune response could be a double-edged sword in the context of the SARS-CoV-2 pandemic [14,15,16,17]. 

In this framework, it is really interesting that patients with rheumatic musculoskeletal disorders (RMD) on treatment with immunosuppressive therapies do not seem to have an increased risk of complications compared to the general population [12,18,19,20,21,22,23]. Even if there is no indication of precautionary suspension of immunosuppressors during the pandemic, since it could lead to disease flares, such treatments should be temporarily stopped in case of active infectious disease, similarly to other infections. On the other hand, the treatment with anti-CD20 agents could lead to increased risk of complications, because of important defects on B-cells response. Therefore, this last group of patients should be more aware of preventing infection [24,25]. More generally, updated results suggest that COVID-19 in patients with inadequate control of inflammatory RMD could be associated with poor prognosis, compared to those in remission [18,19,26]. 

As data comparing RMD patients with the general population are still scarce, the aim of this study is to examine the prevalence of COVID-19 during the first two waves, and the disease patterns of COVID-19 in RMD patients under immunosuppressive treatment.

## 2. Methods

### 2.1. Objective

The primary objective of the present study is to assess the prevalence and the severity of COVID-19 in a population of patients suffering from inflammatory RMD under treatment with a biologic agent or a small molecule during the first two waves, before the vaccination campaign. The second aim is to compare this prevalence with that of the general population (of the region of interest). Finally, the third aim is to describe COVID-19 manifestations among subjects on treatment with biologic DMARDs (b-DMARDs) or small molecules (ts-DMARDs). 

### 2.2. Study Population, Reference Population, and Procedure to Diagnose COVID-19

As described in a previous work [27], the cases were all the adult patients with a rheumatic disease and who were under treatment with a b- or a ts-DMARD (at least from 6 months) from September 2019 to November 2020 in the province of Udine (around 500,000 inhabitants), Italy. Data include the assessment of both pandemic waves: the first has been registered between February 2020 and April 2020; the second one between September 2020 and April 2021. However, in this study, data about the second wave have been considered only until November 2020, as the start of the vaccination made study evaluation more complicated. All the clinical charts of these cases were revised to verify that they were proceeding with treatment at the last contact. The prevalence of COVID-19 during the study period was compared to that of the general population in the province of Udine and in the Friuli-Venezia region Giulia (around 1,200,000 inhabitants) after excluding subjects ≤15 years old [28]. As already specified, all the study patients who undergo a biologic or small molecule treatment must be evaluated by a public consultant rheumatologist every 6 months for renewal and proceeding with their own therapeutic plan, and then they need to be registered by the pharmacy service that supplies the drug about every 2 months until the treatment plan expiration. This aspect ensures updated and reliable data about treatments. Details about the diagnostic tests used for COVID-19 have been described in detail elsewhere [27]. 

The study was conducted in accordance with the ethical principles of the Helsinki Declaration, and approved by the local Ethics Committee (“Comitato Etico Unico Regionale”, CEUR-2020-Os-129).

### 2.3. Statistical Analysis

Results are reported using median (inter-quartile range, IQR) in the case of continuous variables, while categorical variables are described by their absolute frequencies and percentages. 

For univariable comparisons of characteristics between groups, the Wilcoxon–Kruskal–Wallis test is used for continuous variables; regarding the categorical variables, the Pearson’s chi-square test is used [29]. The value of statistical significance considered as possible evidence of a difference between groups, after adjustment of the test values for test multiplicity according to the method by Benjamini and Hochberg, is set as *p* of 0.05 [30].

Analysis has been performed using R software [31] and RMS package [32].

## 3. Results

### 3.1. Epidemiologic Data

Within the province of Udine, 1051 patients taking biologic drugs or small molecules were identified. There were 703 (66.9%) women and 348 (33.1%) men, with a median age of 59 years (IQR 48–70). From 29 February 2019 (the first case of COVID-19 infection reported in Friuli Venezia Giulia) to 30 November 2020, 37 (3.5%) RMD patients resulted positive for COVID-19. In Table 1 cohort characteristics are reported according to COVID-19 test results. No significant differences have been detected between the two groups.

### 3.2. Comparison with Population Data

Overall, disease prevalence among RMD patients was estimated at around 3.6%, with no significant differences compared to the general population of the same province and of the entire region, considering the same period of time. Figure 1 shows the rate of COVID-19 positive cases across regional, provincial, and observed cases, reported overall and stratified by waves. Table 2 reports the disease prevalence and a 2-sample test for equality of proportions with continuity correction, for comparisons between observed rates and those calculated for the province and for the region of interest. The observed rate was significantly different from province rate (*p* < 0.05 for the first wave, *p* < 0.001 for the second wave and overall) but similar to the disease rate recorded in the whole region. 

### 3.3. Characteristics of COVID-19 Patients and Disease Pattern

Table 3 reported the characteristics of the patients who tested positive for COVID-19. Overall, in our province, 37 RMD patients tested positive for COVID-19: 5/37 became infected during the first wave (13.5%), 32/37 in the second one (86.5%). None of the five patients became reinfected in the second wave. About half of the sample was male (46%), the median age was 60 years (IQR 49–68) and the median RMD duration was 6 years (IQR 3–15). The most frequent treatments were represented by anti-TNF (24/37, 65%) and anti-IL17/23 (6/37, 16%).

A small percentage of COVID-19 positive subjects was receiving low doses of corticosteroids (3/37, 8%), while the rest did not receive corticosteroids. Among these patients, 2/3 were taking 5 mg/day and 1/3 were taking 3.5 mg/day of prednisone equivalent. Only a small percentage developed dyspnoea (7/37, 19%) and cough (13/37, 35%). More than 2/3 of the patients developed fever. Among the 6 patients that remained apyretic, 4/6 were on anti-TNF and 1/6 was on anti-IL6 (2 anti-TNF monotherapy, 1 anti-TNF combined with methotrexate, 1 anti-TNF and sulphasalazine, 1 anti-IL6 and methotrexate, 1 on azathioprine and rituximab). Gastro-intestinal symptoms (such as nausea, vomiting, and diarrhoea) have been reported by 10 subjects (27%). Only a fraction of the patients (6/37, 16%) continued ongoing treatment or changed therapy without suspension of immunosuppressants, with no worsened outcomes. Nine patients needed to be hospitalized (24%) and one patient died (carrying comorbidity). After COVID-19 infection, 12 subjects (32%) had RMD flare and 5 of them subsequently needed to change the immunosuppressive therapy, while the other 7 patients did not modify the treatment (in some cases there was a disease flare after the withdrawal of immunosuppressant, with remission of manifestations after restarting the same drug). Among the patients with flare, 5/12 had experienced gastro-intestinal symptoms.

In RMD patients, gastro-intestinal symptoms were associated with longer naso-pharyngeal swab positivity for SARS-CoV-2 (Figure 2). Figure 2 shows through a boxplot the days until negative swab. Although there was no difference between the two COVID-19 waves (Figure 2A), in Figure 2B this phenomenon is described according to the presence of gastro-intestinal symptoms, which identified a subgroup with longer positivity. 

## 4. Discussion

RMD patients are usually considered frail and well known to be at higher risk of severe infections, not only because of their immune dysfunction but mainly for immunosuppressant therapies. Our study shows that the prevalence of COVID-19 in this group of individuals was similar to the general population and those patients did not show a poorer evolution, since the infection generally progressed with a benign trend, in line with other works [18,19,20,21,22,23,26,27,33,34]. 

In fact, even comparing COVID-19 prevalence between the first and the second wave, it could be observed that, despite the greater number of infected, cases with COVID-19 among immunosuppressed patients are statistically lower than among the general population in the same area, in particular in the second wave. It is reasonable to argue that RMD patients have been informed about the potentially higher risk of serious course of COVID-19, and this information has led to improved awareness, i.e., stricter isolation, realization of hygienic measures [27].

Overall, the whole cohort of RMD patients has been firstly described in a previous work [27]. 

Furthermore, our study confirms that immunosuppressant therapy (in particular b- and ts-DMARDs) seems not to be associated with higher risk of COVID-19 complications nor with severe prognosis. These findings are consistent with published data about the safety of the aforementioned therapies [21,35,36,37], particularly referring to the association between anti-TNF treatments, and both the reduced odds of hospitalization and the lower rate of complications during COVID-19, since also in our cohort the most used biologic drugs were anti-TNFs (24/37). Interestingly, the hospitalization rate in our cohort of RMD patients is even lower compared to the study performed by Gianfrancesco et al. (24% vs. 46%, respectively) [21].

Results from this study show that chronic corticosteroid therapy is mostly avoided or, when necessary, its dosage is kept very low (in fact, only 3 out of 37 patients used chronic corticosteroids, at a dosage between 3.5 mg and 5 mg per day in prednisone equivalent). Different surveys agreed on the risk associated with corticosteroid treatment, in particular for medium–high doses and long-term use, since it was associated with increased mortality rate and poor prognosis in COVID-19 patients [18,21,26,37]. 

Another interesting and original result is that COVID-19 gastro-intestinal manifestations are associated with delayed viral clearance, demonstrated by persistent naso-pharyngeal swab positivity for SARS-CoV-2. This aspect suggests a possible role of intestinal microbiota that can be altered in patients with several chronic inflammatory diseases [38,39], identifying a possible predictor of viral persistence in those subjects. Our findings on this particular topic confirmed the available data of another Chinese study that showed the correlation between delayed swab negativization and digestive symptoms, compared to respiratory manifestations [40]. However, the study of Han et al. was performed on the general population, while we included only immunocompromised subjects affected by RMD. Moreover, in our study, persistent swab positivity for COVID-19 and gastro-enteric symptoms were not associated with higher disease severity, since those patients had a benign course and good prognosis. This aspect is in agreement with two other available works [40,41], where the subgroup of COVID-19 patients with digestive symptoms was characterized by low severity infectious disease. Conversely, Ye et al. reported that cases with gastro-intestinal involvement were more likely to be complicated by acute respiratory distress syndrome and liver damage, with poor prognosis [42].

The role of RMD and related therapies on the gut microbiome are currently under study and only preliminary results can be found in literature [39]. Incidentally, especially in COVID-19 patients with gastrointestinal manifestations, gut microbiota may play a role, since imbalance on intestinal flora of COVID-19 patients has been recognized in previous studies, compared with non-COVID-19 patients [43,44].

Nevertheless, the correlation between gastro-intestinal symptoms and longer swab positivity may predict the development of persistent manifestations after recovery (in particular fatigue and arthralgias), due to virus persistence, as observed in many works [6,7,8,39,43,44,45,46,47,48,49,50], but further studies are needed, and longer follow-up of those patients is mandatory. 

Moreover, these data could help clinicians to identify a population of infected subjects among which the early administration of monoclonal antibodies or other specific oral therapies might be useful. 

In terms of SARS-CoV-2 manifestations, our data have shown that patients with rheumatologic diseases developed the same symptoms of general population. Compared to EULAR COVID-19 Registry (as 1 October 2021), our cohort developed a cough in a smaller percentage (respectively, 52% vs. 35%).

Almost all patients in our study cohort developed fever during the infection, despite the anti-cytokine effect of immunosuppressants. No differences were found among patients on treatment with anti-IL6 (tocilizumab or sarilumab), although those therapies sometimes may hide clinical and laboratory manifestations of systemic inflammation in many situations, particularly during other infections, often making difficult or delaying identification in those patients. The small percentage of apyretic individuals was heterogeneous for rheumatologic disease and treatments, excluding possible correlations with them. However, we could observe that, among apyretic patients, 4/6 were treated with anti-TNF while 1/6 with anti-IL6. 

Another possible explanation for the benign prognosis of our cohort is the general low rate of comorbidities, with a median Charlson Comorbidity Index between 0 and 1, as shown in Table 3. In fact, previous Italian studies, including the one performed on data from the Italian Registry of Italian Society of Rheumatology (CONTROL-19), showed that severe prognosis and necessity of intensive care in RMD patients were related to several risk factors shared with the general population, such as older age, male sex, and pre-existing comorbidities [18].

These findings were confirmed in the Euro-COVIMID multi-centre cross-sectional study, that observed the SARS-CoV-2 infection course on rheumatologic patients of six European countries [19]. In all probability, patients on DMARDs are supposed to be more aware on preventive strategies, such as stricter isolation, proper mask use, social distancing, and hand hygiene, and may have less contact with infected people [51]. 

This study has several strengths. First, the same population of RMD patients has been studied in the transition between the first and second wave of COVID-19 pandemic and in a pre-vaccination period. Therefore, our conclusions are not affected by the effect of vaccines nor virus variants. On this subject, the principal strength of that study lies in the fact that current viral variants can overcome vaccine protection and the efficacy of them among immunocompromised patients is still under investigation, so that study has an important value. Second, the studied population was well characterized from a clinical and therapeutic point of view and all subjects were treated in a single centre: this guarantees a homogeneous management both for treatment and follow-up. Then, the most used class of drugs in this cohort is represented by the anti-TNF drugs and this made it possible to study their effect on RMD patients affected by COVID-19. On the other hand, the main limitations of the study are represented by the fact that the sample is not large, and longer follow-up periods are necessary to verify if the long persistence of gastrointestinal symptoms and the delayed viral clearance correlate with further clinical manifestations after recovery. Moreover, misclassification due to diagnosis or hospitalization in a different Italian province may be another possible limitation, however, this possibility is unlikely due to the major restrictions on people circulation through the Italian regions during the first waves of the pandemic.

To conclude, our study reported an overall low prevalence of COVID-19 infection in RMD patients, even if they were under immunosuppressive therapies; this could be due to their high awareness of preventive strategies.

Furthermore, immunosuppressed rheumatologic patients infected by SARS-CoV-2 generally showed mild symptoms and good prognosis. Those findings could be partially explained by the large use of anti-TNF treatments, recognized as a possible protective factor also in other surveys [21,35,36]. Consequently, we should be reassured to continue the ongoing therapy with DMARDs and be more aware for losing RMD control, avoiding the chronic use of corticosteroids if possible. Patients and clinicians should also pay attention to gastrointestinal symptoms to early identify COVID-19 carriers with uncommon symptoms, eligible for the administration of monoclonal antibodies or oral antiviral therapies.

## Figures and Tables

**Figure 1 viruses-14-01462-f001:**
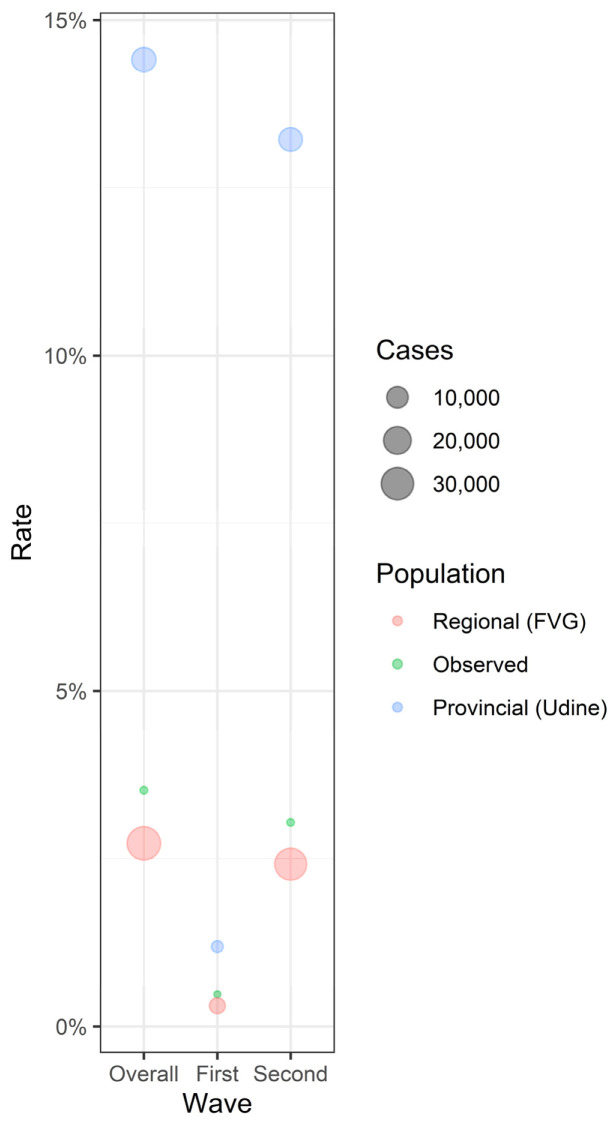
Rate of COVID-19 positive cases across regional (red), provincial (blue), and observed (green) cases, reported overall and stratified by waves.

**Figure 2 viruses-14-01462-f002:**
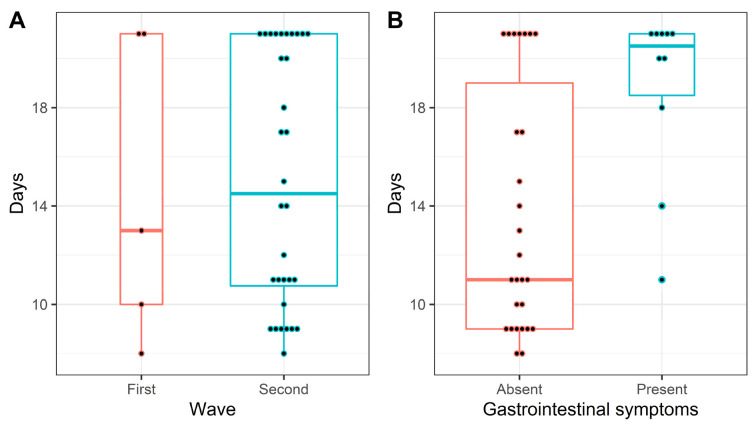
Boxplot and stacked dot-plot (one dot each patient) for days until negative swab across the first two COVID-19 waves (**A**) and according to presence of gastro-intestinal symptoms (**B**). From day 21, everyone is considered as having a negative swab. The boxes report I, II (median), and III quartiles, whiskers extend at most 1.5*Inter-Quartile range from the hinges.

**Table 1 viruses-14-01462-t001:** Characteristics of RMD patients according to COVID-19 test results.

Characteristics	N. of Observations	Negative (*N* = 1014)	Positive (*N* = 37)	Combined (*N* = 1051)	*p*-Value
Age	1051	59 (48–70)	60 (49–69)	59 (48–70)	0.99
Sex (Male)	1051	330 (33%)	18 (49%)	348 (33%)	0.47
RMD					
RA	936	354 (39%)	16 (43%)	370 (40%)	0.89
PsA	937	269 (30%)	10 (27%)	279 (30%)	0.99
SpA	934	167 (19%)	9 (24%)	176 (19%)	0.73
SLE	934	39 (4%)	0 (0%)	39 (4%)	0.73
Vasculitis	934	73 (8%)	2 (5%)	75 (8%)	0.87
Other	928	19 (2%)	0 (0%)	19 (2%)	0.73
b- or ts-DMARD	1051				0.45
Anti-TNF		611 (60%)	24 (65%)	635 (60%)	
JAK inhibitors		69 (7%)	4 (11%)	73 (7%)	
Anti-B cells		66 (7%)	1 (3%)	67 (6%)	
Anti-IL17/23		96 (10%)	6 (16%)	102 (10%)	
Anti-IL6		102 (10%)	2 (5%)	104 (10%)	
Other therapies		70 (7%)	0 (0%)	70 (7%)	
Comorbidities					
Hypertension	914	268 (30%)	11 (32%)	279 (31%)	0.99
Diabetes	914	61 (7%)	4 (12%)	65 (7%)	0.73
Chronic heart disease	915	105 (12%)	4 (12%)	109 (12%)	0.99
History of cancer	913	81 (9%)	5 (15%)	86 (9%)	0.73
Obesity	896	103 (12%)	6 (19%)	109 (12%)	0.73
Duration of last b- or ts-DMARD (months)	878	24 (8–60)	12 (6–36)	24 (8–58)	0.47
Concomitant treatment					
Methotrexate	906	248 (28%)	11 (33%)	259 (29%)	0.87
Corticosteroids	911	143 (16%)	3 (9%)	146 (16%)	0.73
Hydroxychloroquine	910	69 (8%)	1 (3%)	70 (8%)	0.73
Smoke	643	90 (15%)	4 (15%)	94 (15%)	0.99

Data are reported as median (inter-quartile range—IQR) for continuous variables, and absolute frequencies (%) for categorical ones. Legends: RMD = rheumatic musculoskeletal disorders, RA = rheumatoid arthritis, PsA = psoriatic arthritis, SpA = spondyloarthritis, SLE = systemic lupus erythematosus, TNF = tumor necrosis factor, IL = interleukin, b-DMARD = biologic-disease modifying anti-rheumatic drugs, ts-DMARD = targeted synthetic-disease modifying anti-rheumatic drugs. Notes: anti-TNF = adalimumab, infliximab, golimumab, certolizumab, etanercept; JAK-inhibitors = baricitinib, tofacitinib; anti-IL17/23 = secukinumab, ixekizumab, ustekinumab; anti-IL6 = tocilizumab, sarilumab; anti-B cells = rituximab, belimumab; other therapies = abatacept, anakinra, canakinumab, apremilast; corticosteroids = in prednisone equivalents.

**Table 2 viruses-14-01462-t002:** Cases and proportion of COVID-19 among different populations.

	Cases (Proportions)		
Periods	RMD Patients	Province Population	Region Population
Overall wave	37 (0.0352) *	14,437 (0.1441)	32,970 (0.0273)
First wave	5 (0.0048) °	1190 (0.0119)	3769 (0.0031)
Second wave	32 (0.0304) *	13,247 (0.1322)	29,201 (0.0242)

* *p* < 0.001, patients vs. province population; ° *p* < 0.05 patients vs. province population.

**Table 3 viruses-14-01462-t003:** Characteristics of RMD patients suffering from COVID-19.

Characteristics	*N* = 37
Sex (Male)	18 (49%)
Age	60 (49–69)
RMD	
Rheumatoid Arthritis	16 (43%)
Psoriatic Arthritis	10 (27%)
Other Seronegative Spondyloarthritis	9 (24%)
Other	2 (5%)
RMD duration (years)	6 (3–15)
Therapy	
Corticosteroids	3 (9%)
cs-DMARDs	15 (39%)
Anti-TNF	24 (65%)
Anti-IL17/23	6 (16%)
JAK-inhibitors	4 (11%)
Anti-IL6	2 (5%)
Rituximab	1 (3%)
Charlson Comorbidity Index	0 (0–1)
Wave of COVID-19	
First	5 (14%)
Second	32 (86%)
Symptoms	
Fever	26 (70%)
Fatigue	22 (59%)
Myalgia	18 (49%)
Arthralgia	20 (54%)
Cough	13 (35%)
Dyspnoea	7 (19%)
GI symptoms	10 (27%)
Anosmia	10 (27%)
Ageusia	10 (27%)
Disease course	
Hospitalization	9 (24%)
ICU admission	2 (5%)
Disease flare after COVID-19	12 (32%)

Data are reported as median (inter-quartile range—IQR) for continuous variables and absolute frequencies (%) for categorical ones. Legend: RMD = rheumatic musculoskeletal disorders; cs-DMARDs = conventional synthetic Disease Modifying Anti-Rheumatic Drugs; GI, gastro-intestinal symptoms; ICU, intensive care unit.

## Data Availability

The data presented in this study are available on request from the corresponding author. The data are not publicly available due to the use of local administrative resources.

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
