# Peer review of "Safety of Biologic-DMARDs in Rheumatic Musculoskeletal Disorders: A Population-Based Study over the First Two Waves of COVID-19 Outbreak"

_viruses, 2022, doi:10.3390/v14071462_

Round 1

Reviewer 1 Report

Scientific sound publication.

Two minor ooments according to the discussion: 

a. RMD patients have "a priori" been informed about the potentially higher risk of serious course of Covid 19. There is reasonable evidence for the fact, that this information has led to improved awareness, i.e. more consequent isolation. relisaton of hygienic concepts.  i would advise to discuss these factors.

b. The articel states that "Another possible explanation for the benign prognosis of our cohort is the general low rate of comorbidities, with a median Charlson Comorbidity Index between 0-1-"

Likewise, it can be discussed if especially those RMD patients, who were a priori aware of their older age/ well known comorbidities used more explicit and intensive isolation/ hygienic strategies to avoid infection in first place

Reviewer 2 Report

The authors have a low rate of COVID-19 in their RMD cohort. If the patients were diagnosed or hospitalized in another Italian city could they be misclassified in your report? If so, please include this in your limitations and discussion.

Please correct sentence “Infected population could present a wide spectrum of manifestations and sometimes none symptom, making difficult an early diagnosis and facilitating virus circulation” to increase clarity.

Please include some sentences (in the second paragraph of the discussion section) discussing the potential reasons that may explain why patients with RMD has a lower prevalence of COVID-19 (stricter isolation or hygiene measures, bias in testing?).

Only 5 cases had Covid in the first wave and 32 cases occurred in the second wave of Covid. Were there any of those 5 cases of infected patients in the first wave that became reinfected?
